# Alleviation of Hyperuricemia by Strictinin in AML12 Mouse Hepatocytes Treated with Xanthine and in Mice Treated with Potassium Oxonate

**DOI:** 10.3390/biology12020329

**Published:** 2023-02-17

**Authors:** Kuo-Ching Huang, Yu-Ting Chang, Rosita Pranata, Yung-Hsuan Cheng, Yu-Chi Chen, Ping-Chung Kuo, Yi-Hsuan Huang, Jason T. C. Tzen, Rong-Jane Chen

**Affiliations:** 1Division of Nephrology, Department of Internal Medicine, Chi Mei Hospital, Tainan 736, Taiwan; 2Department of Environmental and Occupational Health, College of Medicine, National Cheng Kung University, Tainan 701, Taiwan; 3Department of Food Safety/Hygiene and Risk Management, College of Medicine, National Cheng Kung University, Tainan 701, Taiwan; 4Department of Biotechnology, National Kaohsiung Normal University, Kaohsiung 824, Taiwan; 5School of Pharmacy, College of Medicine, National Cheng Kung University, Tainan 701, Taiwan; 6Graduate Institute of Biotechnology, National Chung Hsing University, Taichung 402, Taiwan

**Keywords:** hyperuricemia, NLRP3 inflammasome, strictinin, uric acid, xanthine

## Abstract

**Simple Summary:**

Hyperuricemia is a major risk factor for gout. Inhibition of liver xanthine oxidase has been shown to reduce the uric acid level in blood. However, side effects were reported for the xanthine oxidase inhibitors clinically used in the treatment of hyperuricemia. This study aimed to evaluate strictinin, a natural polyphenol from Pu’er tea, for its therapeutic effects on hyperuricemia. Alleviation of hyperuricemia by strictinin supplementation was observed in AML12 mouse hepatocytes treated with xanthine as well as in an animal model using mice treated with potassium oxonate.

**Abstract:**

Hyperuricemia, an abnormally high level of blood uric acid, is a major risk factor for gout. Although xanthine oxidase inhibitors were clinically used to lower blood uric acid level, the concerned side effects restricted their utilization. In this study, strictinin, an abundant polyphenol in Pu’er tea, was evaluated for its preventive effects on hyperuricemia. The results showed that the xanthine oxidase activity, uric acid production, and inflammation in AML12 mouse hepatocytes treated with xanthine were significantly reduced by the supplementation of strictinin. Detailed analyses revealed that strictinin inhibited xanthine-induced NLRP3 inflammasome activation. Consistently, the elevated blood uric acid level and the enhanced xanthine oxidase activity in mice treated with potassium oxonate were effectively diminished by strictinin supplementation. Moreover, for the first time, strictinin was found to promote healthy gut microbiota. Overall, strictinin possesses a great potential to be utilized as a functional ingredient for the prevention of hyperuricemia.

## 1. Introduction

Hyperuricemia is a metabolic disease characterized by high blood uric acid (UA). It results from the excess UA synthesized in the liver and/or incompetent excretion from the kidney. In Taiwan, the proportions of men and women with their blood uric acid levels exceeding 7.0 mg/dL were reported to be 46% and 26%, respectively [1]. The high level of UA is a typical impediment in tubules and may lead to the progression of hyperuricemic nephropathy, as shown by the deposition of UA crystals enclosed by macrophages in the kidneys of hyperuricemic animals [2].

Hyperuricemia is noticed as a major risk factor for gout and metabolic syndromes, including several chronic diseases in the liver, heart, and kidney [3,4,5]. High dietary intake of high-purine foodstuffs, such as alcoholic beverages, organ meats, seafood, and shellfish, might increase the UA level in blood. Purine (guanosine and inosine) nucleotides from diets are converted to xanthine by way of enzymatic reactions in the liver putatively by specific enzymes including nucleotidase, purine nucleoside phosphorylase, guanine deaminase, and xanthine oxidase (XOD) [6]. In the human body, xanthine is assumed to be oxidized to UA by XOD, a highly expressed enzyme in the liver [7]. As a result, reducing UA production by modulating liver enzymes responsible for UA production is assumed to effectively prevent the UA-induced metabolic diseases.

Recent studies showed that NOD-like receptor family pyrin domain containing 3 (NLRP3) inflammasome presumably played a key role in the development of many diseases, including UA-induced kidney inflammation and fibrosis [6]. NLRP3 inflammasome contains several proteins, such as NLRP3, pro-caspase-1, and the speck-like protein with a caspase recruitment domain (ASC) [8,9]. Activation of NLRP3 inflammasome requires two steps. The first step is the initiation reaction, where the cells are subjected to pathogen-associated molecular patterns (PAMPs) or damage-associated molecular patterns (DAMPs). NF-κB signal transduction is activated to promote the production of NLRP3, pro-IL-1β, and pro-IL-18. The second step is the activation reaction, where caspase-1 triggers the generation of IL-1β and IL-18 to promote the inflammation response. Notably, UA, a DAMP, was shown to induce the production of reactive oxygen species (ROS) and further NLRP3 inflammasome activation [6]. In addition, activation of XOD seemed to activate the NLRP3 inflammasome [10]. Therefore, inhibition of XOD activity might be beneficial to reduce UA production and NLRP3 inflammasome activation [11].

Allopurinol, an inhibitor of XOD, is clinically applied to the treatment of hyperuricemia and gout by stimulating the renal excretion of UA [12]. However, side effects such as allergies, skin rash, liver necrosis, and poor renal function were reported for this drug [13]. Thus, it is essential to develop new drugs without severe side effects for the treatment of hyperuricemia. Strictinin, a hydrolysable tannin abundantly found in Pu’er tea, was demonstrated to display a broad range of biological functions, such as anti-obesity, anti-tumor, antipsoriatic, anti-microbial, and anti-viral activities [14,15,16,17,18]. In this study, the preventive effects of strictinin on hyperuricemia were investigated in a cellular model as well as in an animal model. Moreover, gut dysbiosis has been suggested to play an important role in the pathogenesis of inflammation and metabolic disease, and Pu’er tea was known to possess cholesterol- and lipid-lowering effects through modulation of gut microbiota [19,20]. Hence, whether the UA-lowering effect of strictinin could be partially attributed to modulation of intestinal microbiota was also evaluated.

## 2. Materials and Methods

### 2.1. Chemicals and Reagents

Strictinin was purified from Pu’er tea according to the protocol developed previously [14]. Potassium oxonate (uricase inhibitor) and allopurinol (XOD inhibitor) were bought from Sigma-Aldrich (St. Louis, MO, USA). MCC950 (NLRP3 inhibitor) was obtained from Cayman Chemical (Ann Arbor, MI, USA). Primary antibodies against NLRP3, pro-caspase-1, and ASC were purchased from Abcam (Cambridge, MA, USA). Primary antibodies against caspase-1 and JNK were purchased from Epitomics (Cambridge, MA, USA). Primary antibody against IL-1β was purchased from PeproTech (Rocky Hill, NJ, USA). Primary antibodies against GAPDH, p-NF-κB, NF-κB, p-ERK1/2, ERK1/2, and p-JNK as well as horseradish peroxidase-conjugated anti-mouse and anti-rabbit secondary antibodies were supplied by the Cell Signaling Technology (Beverly, MA, USA).

### 2.2. Cell Culture

Mouse hepatic cell line AML12 was supplied by the American Type Culture Collection (ATCC^®^ CRL-2254). The cells were amplified in DMEM/F-12 (Gibco; Thermo Fisher Scientific, Inc., Billings, MT, USA) with the supplementation of 10% fetal bovine serum (Gibco), 100 µg/mL of penicillin and streptomycin, 5 µg/mL of ITS-M (Simply Biologics, Miaoli, Taiwan), and 40 ng/mL of dexamethasone (Sigma-Aldrich, Darmstadt, Germany). The cultured cells were incubated under the atmosphere of 5% CO_2_ at 37 °C, and then plated on a 10 cm dish (approximately 800,000 cells per plate) for the following experiments.

### 2.3. Cell Viability Assay

The MTT (3-[4,5-dimethylthiazol-2-yl]-2,5-diphenyl tetrazolium bromide) assay was employed to assess cytotoxicity. AML12 cells were loaded into 96-well plates (approximately 8000 cells per well). After incubation with strictinin concentrations of 50, 100, 250, or 500 μM for 24 h, the cells were rinsed with phosphate buffered saline (PBS). Thereafter, 100 μL of 10% MTT solution was applied to the wells and they were further incubated for 2 h at 37 °C. The MTT solution was decanted after incubation, and 100 μL of dimethyl sulfoxide (DMSO, Sigma-Aldrich) was supplied to the wells to dissolve the crystals. An enzyme-linked immunoassay (ELISA) analyzer (Cayman Chemical, Ann Arbor, MI, USA) was employed to detect the optical density of each well at a wavelength of 595 nm. All experiments were executed in triplicates.

### 2.4. Detection the Production of UA and IL-1β

AML12 cells were rinsed with PBS and then treated with xanthine (UA precursor) of 100 or 200 µM with or without strictinin of 100 or 250 µM. After incubation for 4 and 24 h, 100 µL of medium was taken to determine the concentration of UA using the Uric Acid Assay Kit following the producer’s protocol (Cayman Chemical, Ann Arbor, MI, USA). The content of IL-1β in the supernatant was measured by a commercial IL-1β Mouse ELISA Kit (Invitrogen, Thermo Fisher Scientific, Inc., Waltham, MA, USA). All tests were executed in triplicates according to the manufacturer’s instructions.

### 2.5. Western Blotting Analysis

Cell lysates were collected by centrifugation (10,000× *g*) at 4 °C for 30 min. After centrifugation, the proteins in the supernatant were subjected to separation by sodium dodecylsulfate polyacrylamide gel electrophoresis, and transferred onto a polyvinylidene difluoride membrane. The membrane was immersed in 5% non-fat milk for 1 h prior to incubation with the primary antibodies at 4 °C overnight. Primary antibodies against GAPDH (1:5000), NLRP3 (1:1000), caspase-1 (1:1000), pro-caspase-1 (1:1000), ASC (1:1000), p-NF-κB (1:1000), NF-κB (1:1000), IL-1β (1:1000), p-ERK1/2 (1:1000), ERK1/2 (1:1000), p-JNK (1:1000), and JNK (1:1000) were used in the immunological detection. Membranes were rinsed with Tris-buffered saline containing 0.1% Tween^®^ 20 (TBST) 3 times prior to incubation with secondary antibodies at 37 °C for 1 h. Then, the target protein bands were imaged using the enhanced chemiluminescent reagent (Invitrogen) and quantitated by the iBright Imaging Systems (iBright FL 1000; Thermo Fisher Scientific, Inc., Waltham, MA, USA).

### 2.6. In Vivo Experiment

8-week-old male ICR mice (BioLASCO Taiwan Co., Ltd., Taipei, Taiwan) were allowed to freely access to tap water and the regular rodent diet following the guidelines for animal protection. They were housed in plastic cages within the Laboratory Animal Center of the National Cheng Kung University Medical College under a 12 h/12 h light and dark cycle in a pathogen-free room at 24 ± 2 °C and 50 ± 10% relative humidity with an approval no. 108240. The mice were allocated into six groups with comparable body weight (b.w.) in each group (n = 3); (1) control group: normal diet without any treatment, (2) hyperuricemic group: administered with potassium oxonate (PO) of 400 mg/kg b.w., (3) positive control group: PO of 400 mg/kg b.w. + allopurinol (AP) of 10 mg/kg b.w., (4) strictinin-400 group: PO of 400 mg/kg b.w. + strictinin (ST) of 400 mg/kg b.w., (5) strictinin-700 group: PO of 400 mg/kg b.w. + ST of 700 mg/kg b.w., and (6) strictinin-1000 group: PO of 400 mg/kg b.w. + ST of 1000 mg/kg b.w. PO and AP were suspended in saline while ST was in water. After fasting for 1 h, PO was given to mice by oral gavage followed by AP or various concentrations of ST approximately 1 h later. The experiment was conducted for 7 days and all animals were sacrificed on the last day. Samples of blood and organs were collected for further analyses, including detection of uric acid level and XOD activity as well as histological staining.

### 2.7. Detection of XOD Activity

XOD activity in serum or liver was determined using the Xantine Oxidase Fluorometric Assay Kit (Cayman Chemical, Ann Arbor, MI, USA) following the producer’s protocol. Serum samples were diluted with the XOD sample buffer. Liver tissue was extracted with the lysis buffer, and the supernatant after centrifugation was collected for analysis. Each sample of 50 μL was loaded into a well in the 96-well plates, mixed with the assay cocktail, and incubated at 37 °C for 45 min. The fluorescence was measured with excitation/emission wavelengths of 540/595 nm and the OD value was calculated for the final concentration.

### 2.8. Histopathological Analysis

Kidney and liver samples were fixed directly in 3.7% formalin for 7 days at room temperature. Serial sections (5 μm in thickness) of kidney and liver were dehydrated through different dilutions of ethanol, and then embedded in paraffin. Prior to hematoxylin and eosin staining, the sections were hydrated with ethanol gradient and washed with tap water. The morphology of renal cells and hepatocytes were observed under a light microscope, the Olympus CK40 (Olympus, Tokyo, Japan).

### 2.9. Analysis of Microbiota

The feces were collected from mice after sacrifice, and sent to the Tri-I Biotech Inc. (Taipei, Taiwan) for analysis of gut microbiota. Microbial DNA fragments were first extracted with the QIAamp Fast DNA Stool Mini Kit (Qiagen, Hilden, Germany). Subsequently, the fragments corresponding to the V3–V4 hypervariable region of 16S rDNA were amplified with 341F and 805R primers. The PCR amplified fragments were isolated by using the QIAquick PCR Purification Kit (Qiagen). After amplification, DNA fragments were collected and further purified by the AMPure XP beads (Beckman Coulter, Indianapolis, IN, USA) and the MinElute Gel Extraction Kit (Qiagen). Finally, a library was constructed with the Celero™ DNA-Seq System (1–96) (NuGEN, Redwood City, CA, USA). Sequencing data were obtained by using the Illumina MiSeq™ System (Illumina Inc., San Diego, CA, USA). Sequences with similarity higher than 97% were classified into identical operational taxonomic units.

### 2.10. Statistical Analysis

The in vivo and in vitro data are expressed as the mean ± standard deviation (SD). Student’s t-test was used for the experimental data to compare the differences between two groups. When the *p*-value was <0.05, the differences were considered statistically significant. The experiments were performed at least three times.

## 3. Results

### 3.1. Effects of Strictinin on XOD Activity, UA Production, and IL-1β Expression in AML12 Cells Treated with Xanthine

Cytotoxicity of strictinin (50, 100, 250, and 500 μM) to AML12 cells was examined first and the results showed that cell viability was significantly reduced when strictinin was supplemented at a concentration of 500 μM (Figure 1A). Therefore, strictinin of 100 and 250 μM was used in the following detection process. As shown in Figure 1B, the XOD activity in AML12 cells treated with xanthine (100 or 200 μM) for 4 h was significantly elevated in comparison with the control group, and the significant elevation of XOD activity was substantially attenuated when strictinin (100 or 250 μM) was supplemented. Similarly, UA production in AML12 cells treated with xanthine for 4 or 24 h was significantly enhanced, and the enhancement of UA production was mostly abolished when strictinin was supplemented (Figure 1C). Inflammation was observed in AML12 cells treated with xanthine of 100 μM for 4 or 24 h as the IL-1β expression was detected, and the xanthine-induced inflammation was significantly inhibited when strictinin of 250 μM was supplemented (Figure 1D).

### 3.2. Inhibition of Strictinin on the ERK1/2, JNK, and Xanthine-Induced NLRP3 Inflammasome Activation

XOD activity might lead to the activation of NLRP3 inflammasome, a modulator involved in various inflammation pathways, and ERK and JNK have been shown to play key roles in signal transduction in response to inflammation [6,21]. As shown in Figure 2A, JNK instead of ERK was activated after xanthine treatment, while strictinin supplement successfully inhibited the activation of ERK and JNK. Furthermore, xanthine treatment significantly enhanced the activation of NF-κB while strictinin supplement decreased the expression of NF-κB. Additionally, xanthine treatment for 6 h significantly upregulated the expressions of NLPR3, ASC, caspase-1, and cleaved caspase-1; the elevated expressions of these NLRP3 inflammasome components were reduced when strictinin was supplemented (Figure 2B).

To further clarify whether the anti-inflammatory effect of strictinin was indeed regulated through inhibition of NLRP3 inflammasome activation, AML12 cells were treated with MCC950, an NLRP3 inflammasome inhibitor, alone or in combination with xanthine. Similar to the results of strictinin supplementation (Figure 2), treatment using MCC950 (1 μM) decreased the activation of ERK, JNK, and NF-κB (Figure 3A), and reduced the expressions of NLPR3, ASC, caspase-1, and cleaved caspase-1 (Figure 3B). Next, caspase-1-silenced AML12 cells were used to confirm that strictinin inhibited inflammation through downregulation of NLRP3 inflammasome activation. After silencing caspase-1, the expression of cleaved IL-1β was significantly decreased, particularly in the strictinin-supplemented groups (Figure 3C). Strictinin seemed to play an important role in inhibiting XOD activity, NLRP3 inflammasome activation, and UA production in hepatocytes.

### 3.3. UA Lowering and XOD Inhibition of Strictinin in Mice Treated with Potassium Oxonate

A hyperuricemia animal model using mice treated with potassium oxonate (PO), a uricase inhibitor, was employed to evaluate the beneficial effects of strictinin in vivo. The body weight of mice treated with PO and allopurinol was found to increase gradually while no significant change was observed for the body weight of mice in the other groups (Figure 4A). No significant differences were detected for the function parameters of liver and kidney, including values of glutamic oxaloacetic transaminase (GOT), glutamic pyruvic transaminase (GPT) (C), and creatinine (CRE) (Figure 4B–D). The blood urea nitrogen (BUN) value was slightly different among groups but not statistically significant (Figure 4E). The liver XOD activity in PO-treated mice was elevated in comparison with the control group, and the elevation was significantly reduced when mice were supplemented with allopurinol or strictinin (Figure 4F). Similarly, the serum UA level in PO-treated mice was elevated in comparison with the control group, and the elevation was significantly reduced when mice were supplemented with allopurinol or strictinin (Figure 4G). The results were in agreement with the observation in the cellular model (Figure 1), indicating that strictinin inhibited liver UA production and XOD activity. Histopathological assessment revealed that no significant damage was observed in the liver tissues of mice from all the groups, and that slight damage with tubular dilation was observed in the kidney tissues of PO-treated mice while no significant damage was observed in the kidney tissues of PO-treated mice supplemented with strictinin (Figure 4H). It seemed that the slight damage in the kidney caused by PO treatment could be rescued by the supplementation of strictinin.

### 3.4. Beneficial Effects of Strictinin on the Gut Microbiota

The impacts of strictinin on gut microbiota of mice after ingestion were evaluated. The taxonomy profiling results showed that no significant difference of Firmicutes/Bacteroidetes (F/B) ratio was observed among the animal groups (Figure 5A). Furthermore, the Venn diagram results showed that 1579 species were shared between the PO-treated group and the control group, 1586 species between the strictinin-treated group and the control group, and 1536 species between the PO+strictinin-treated group and the PO-treated group (Figure 5B). The difference of bacteria between two groups was further analyzed using Linear Discriminant Analysis (LDA). Significant change in *Porphyromonadaceae*, *Psychrophilus*, *Jeotgalicoccus*, *Staphylococcaceae*, *Bacillales*, and *Bacilli* was detected between the PO-treated group and the control group (Figure 5C). Significant change in *Psychrophilus*, *Jeotgalicoccus*, *Eqorem*, *Staphylococcus*, *Staphylococcaceae*, and *Bacillales* was detected between the strictinin-treated group and the control group (Figure 5D). Significant change in *Psychrophilus*, *Jeotgalicoccus*, *Staphylococcaceae*, *Bacillales*, and *Staphylococcaceae* was detected between the PO+strictinin-treated group and the PO-treated group (Figure 5E).

Detailed analysis of bacterial species with a significant difference between the PO-treated group and the control group showed that the proportions of *Clostridium aldenense*, *Clostridium lavalense*, *Clostridium septicum*, *Clostridium saccharolyticum*, and *Butyricicoccus pullicaecorum* were significantly elevated, while those of *Clostridium thermosuccinogenes*, *Marvinbryantia formatexigens*, and *Ruminococcus lactaris* were significantly reduced (Figure 6A). When the strictinin-treated group was compared with the control group, the proportions of *Clostridium clostridioforme*, *Proteocatella sphenisci*, and *Ruminococcus lactaris* were significantly elevated, while those of *Clostridium saccharolyticum*, *Clostridium cellulovorans*, *Clostridium bolteae*, *Clostridium symbiosum*, *Clostridium lavalense*, *Clostridium saccharolyticum*, and *Staphylococcus sciuri* were significantly reduced (Figure 6B). When the PO+strictinin-treated group was compared with the PO-treated group, the proportions of *Clostridium thermosuccinogenes*, *Marvinbryantia formatexigens*, and *Ruminococcus lactaris* were significantly elevated, while those of *Clostridium aldenense*, *Clostridium cellulovorans*, *Clostridium lavalense*, *Clostridium saccharolyticum*, *Clostridium symbiosum*, *Ruminococcus gauvreauii*, *Roseburia faecis*, and *Ruminococcus gnavus* were significantly reduced (Figure 6C).

## 4. Discussion

Hyperuricemia is a condition where the UA level in a patient’s blood is abnormally elevated, clinically defined as >7 mg/dL (420 μM) in men and >6 mg/dL (360 μM) in women [22]. UA of high concentration tends to precipitate in body tissues or body fluids; therefore, persistent hyperuricemia presumably leads to the formation of urate crystals, a risk factor for various metabolic disease [6,23]. XOD is a key enzyme involved in the conversion of xanthine and hypoxanthine to UA in the liver [24]. The increase of XOD activity may lead to the over-synthesis of UA, and thus is also regarded as an indicator of hyperuricemia [25]. Clinically, UA-lowering (hypouricemic) drugs effectively alleviate the painful hyperuricemia-induced symptoms; however, the commonly used hypouricemic drugs, such as allopurinol and febuxostat, have been reported to cause renal and gastrointestinal toxicity, kidney and liver damage, and myelosuppression [26,27]. Searching for potential hypouricemic compounds from natural sources might be an adequate approach to identify new hypouricemic drugs. In this study, strictinin, an antioxidant found in Pu’er tea [28,29], was examined in the established xanthine-treated hepatocyte model and in the PO-induced hyperuricemia animal model. The results indicated that strictinin effectively reduced UA production by inhibiting XOD activity according to both in vitro and in vivo studies. The data suggest that strictinin is a potential hypouricemic agent.

Xanthine has been reported to induce inflammation through the NLRP3 pathway [6]; hence, the anti-inflammatory effects of strictinin against the NLRP3 activation in AML12 cells were evaluated. For the first time, the urate-lowering effect of strictinin was attributed to its anti-inflammation via inactivation of the NLRP3 pathway in the cell model. According to our results, treatment with strictinin was found to inhibit the expression of NLRP3, caspase-1, and IL-1β (Figure 3). IL-1 is known to be a sign of inflammation and is related to cancer and tumor development [30]. A previous study showed that strictinin treatment modulated the PI3K/AKT/GSK3β pathway in triple negative breast cancer [31] and blocked IL-8 mRNA expression in normal human epidermal keratinocytes [32]. In this study, strictinin was shown to block the activation of the ERK1/2, JNK, and NF-κB pathways, which have been noticed to be inflammation-related. The NF-κB pathway has been reported to be upstream of IL-1β and NLRP3 [30], and is now recognized as an inflammation pathway involved in various diseases including urate nephropathy [6].

Our present results also showed that xanthine activated the NLRP3 inflammasome pathway, while strictinin inhibited the NLRP3 inflammasome activation (Figure 3). Notably, silenced caspase-1 expression in AML12 cells significantly reduced IL-1β production, confirming that strictinin inhibited inflammation through the downregulation of NLRP3 inflammasome activation. It is consistent with our previous study showing that pterostilbene, a natural stilbene compound found in berries and grapes, prevented the high-adenine-induced urate nephropathy. This was characterized by renal inflammation and fibrosis, and mainly regulated by TGF-β production and NLRP3 inflammasome pathway activation [6]. Therefore, we strongly suggest that xanthine can induce hyperuricemia in the kidney and inflammation in liver through the activation of NF-κB/NLRP3 inflammasome pathway; whereas strictinin supplementation can reduce UA production and inflammation through modulating NLRP3 inflammasome pathway. Moreover, it has been shown that xanthine treatment activated xanthine oxidase, a key enzyme involved in the production of reactive oxygen species (ROS), and thus might induce ROS production indirectly [33]. Since strictinin is also known to possess anti-oxidative effects [29], it is also possible that the ROS production induced by xanthine treatment can be inhibited by the strictinin supplementation.

The human gut microbiota is composed of thousands of bacterial species. Gut microbiota is also known as the “forgotten organ” due to its vital contribution to human health [34]. In a previous study, significant changes were observed in the gut microbiota of high-purine-induced hyperuricemia rats, suggesting that renal diseases might also result in an imbalance of gut microbiota [35]. In this study, a relatively high abundance of several species including *Clostridium septicum* was detected in PO-treated mice in comparison with the control group (Figure 6). *Clostridium septicum* has been shown to cause aortic aneurysms, which were associated with high mortality and malignancy of colon cancer [36]. Meanwhile, the abundance of *Clostridium thermosuccinogenes*, an attractive production organism for the metabolism of organic acids and succinic acid from lignocellulosic biomass-derived sugars, was reduced in the gut microbiota of PO-treated mice in comparison with the control group. The results suggested that PO-treated hyperuricemic mice might have a relatively high abundance of gut microbiota related to inflammation.

*Clostridium* was reported to be prevalent in obese patients with various metabolic disorders [37] and in obese children with asymptomatic hyperuricemia [38]. In this study, the proportions of some *Clostridium* species including *Clostridium aldenense* [39], *Clostridium lavalense* [40], *Clostridium cellobioparum* [41], and *Clostridium symbiosum* [42], which have been reported to be related to infection and inflammation, were significantly reduced in the gut microbiota of mice treated with strictinin (Figure 6B,C). The result seemed to be in agreement with a previous study showing that strictinin isolated from Pu’er tea possessed anti-obesity effects [43]. In addition, the abundance of *Staphylococcus sciuri,* which has been shown to contribute to autoimmune diseases and anti-microbial resistance [44], was significantly reduced in the gut microbiota of strictinin-treated mice in comparison with the control group (Figure 6B). The results suggested that strictinin might reduce inflammation and infection via modulation of gut microbiota. This suggestion was consistent with previous reports showing that Pu’er tea extracts could alleviate intestinal inflammation and dextran sulfate sodium induced colitis in mice by regulating gut microbiota [45,46].

Recent studies indicated that some species of *Ruminococcus* might be associated with Crohn’s disease, the most common subtype of inflammatory bowel disease [47]. *Roseburia* spp. were reported to affect various metabolic pathways, and were assumed to be associated with several diseases, including irritable bowel syndrome, obesity, type 2 diabetes, nervous system conditions, and allergies [48]. In this study, the proportions of *Ruminococcus gnavus*, *Ruminococcus gauvreauii*, and *Roseburia faecis* were significantly reduced in the gut microbiota of PO+strictinin-treated mice in comparison with those in the gut microbiota of PO-treated mice (Figure 6C). Taken together, the results suggested that hyperuricemia might alter the gut microbiota that was linked to intestinal inflammation, and that strictinin supplementation significantly changed multiple bacterial species that could promote the development of a healthy gut microbiota with protective potential against inflammatory-related diseases. However, most of the phyla and genus are still left unclassified, which calls for further investigation for the effects of strictinin on the modulation of these genera.

The content of strictinin in the dry weight of Pu’er tea ranges from 2 to 10% [14,15]. Daily consumption of tea infusion from 10 g Pu’er tea is commonly acceptable for tea drinkers. Therefore, daily uptake of strictinin up to 500–1000 mg seems to be safe according to the empirical drinking behavior. However, safety issues for taking strictinin of a high dosage as a drug has not been addressed so far. Notably, cell viability of AML12 mouse hepatocytes was dose-dependently reduced by strictinin concentrations of 50–500 μM, and the reduction was found statistically significant when the strictinin concentration was 500 μM (Figure 1A). Moreover, NLRP3 and Caspase-1 were slightly upregulated by strictinin supplementation (100 or 250 μM) though the upregulation was not statistically significant (Figure 2). Taken together, one should be cautious for the development of strictinin as a drug or functional supplement. Definitively, more evaluation, such as toxicity of strictinin in high dosages, should be executed prior to its practical utilization.

## 5. Conclusions

In this study, alleviation of hyperuricemia by strictinin supplementation was observed in AML12 mouse hepatocytes treated with xanthine. It seemed that xanthine induced UA production and inflammation in hepatocytes through the NF-κB/NLRP3 inflammasome pathway, and that strictinin supplementation played a role in the inhibition of inflammation-related pathways, such as ERK1/2, JNK, NF-κB, and NLRP3 inflammasome activation. Consistently, the urate lowering effect, renal protection, and the beneficial effects of strictinin on the development of gut microbiota were observed in an animal model using mice treated with potassium oxonate. It is suggested that strictinin is not only a protective agent to reduce UA production, inflammation, and renal damage but also a health ingredient to improve gut microbiota composition.

## Figures and Tables

**Figure 1 biology-12-00329-f001:**
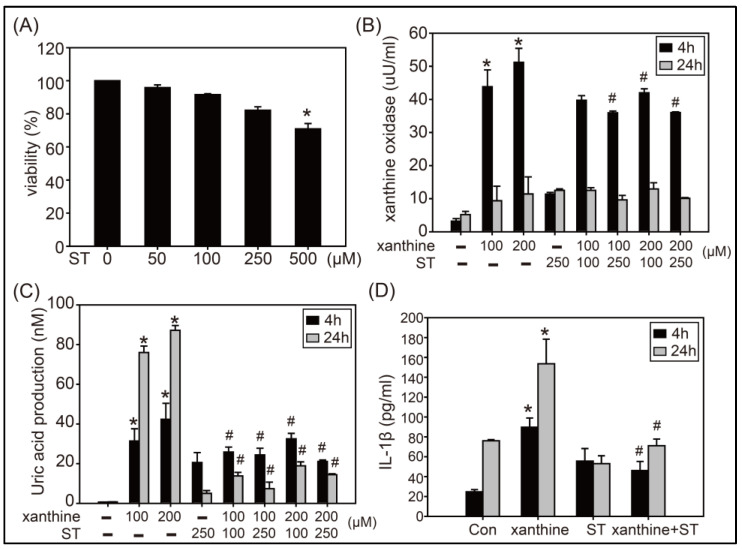
Effects of strictinin (ST) on cell viability (**A**), XOD activity (**B**), UA production (**C**), and IL-1β expression (**D**) of AML12 cells. Cell viability of AML12 cells was examined in the presence of ST concentrations of 50, 100, 250, and 500 μM. Quantitative assays of XOD activity, UA production, and IL-1β expression were performed when AML12 cells were treated with xanthine (100 or 200 µM) and/or ST (100 or 250 μM) for 4 or 24 h. Data are presented as mean ± SD from three independent experiments. * *p* < 0.05 compared with control. # *p* < 0.05 compared with xanthine alone groups.

**Figure 2 biology-12-00329-f002:**
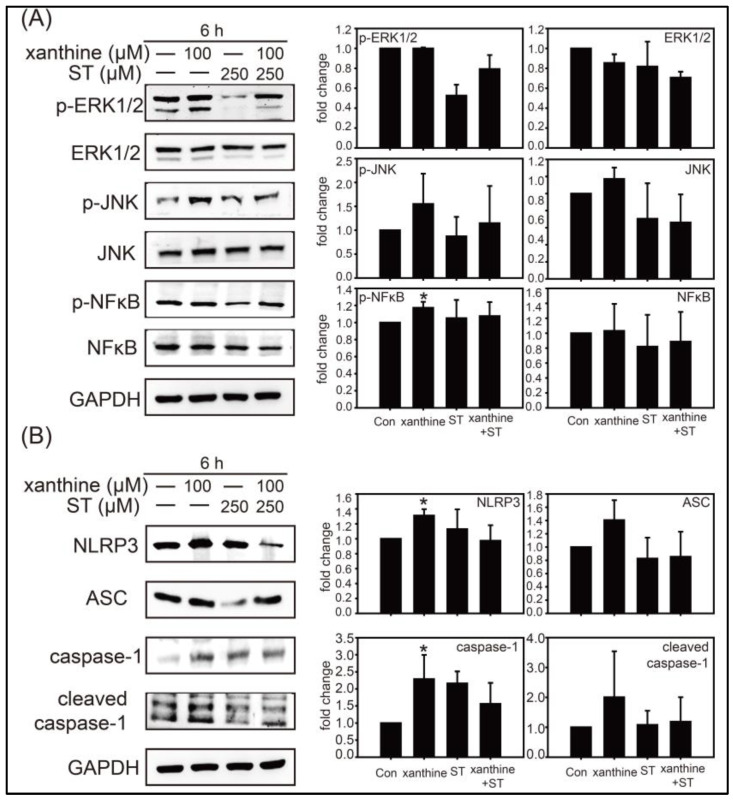
Effects of xanthine and strictinin (ST) on signal transduction and NLRP3 inflammasome pathways in AML12 cells. Western blot analysis was conducted with cell lysates from AML12 cells after treatment with xanthine (100 μM) and/or ST (250 μM) for 6 h. Representative images and quantitation of the Western blot are shown for the protein expressions related to signal transduction pathway (**A**) and NLRP3 inflammasome pathway (**B**). GAPDH was an internal control for Western blot analysis. Data are presented as mean ± SD from three independent experiments. The densitometry readings of Western blot analysis was shown in Appendix A. * *p* < 0.05 compared with control.

**Figure 3 biology-12-00329-f003:**
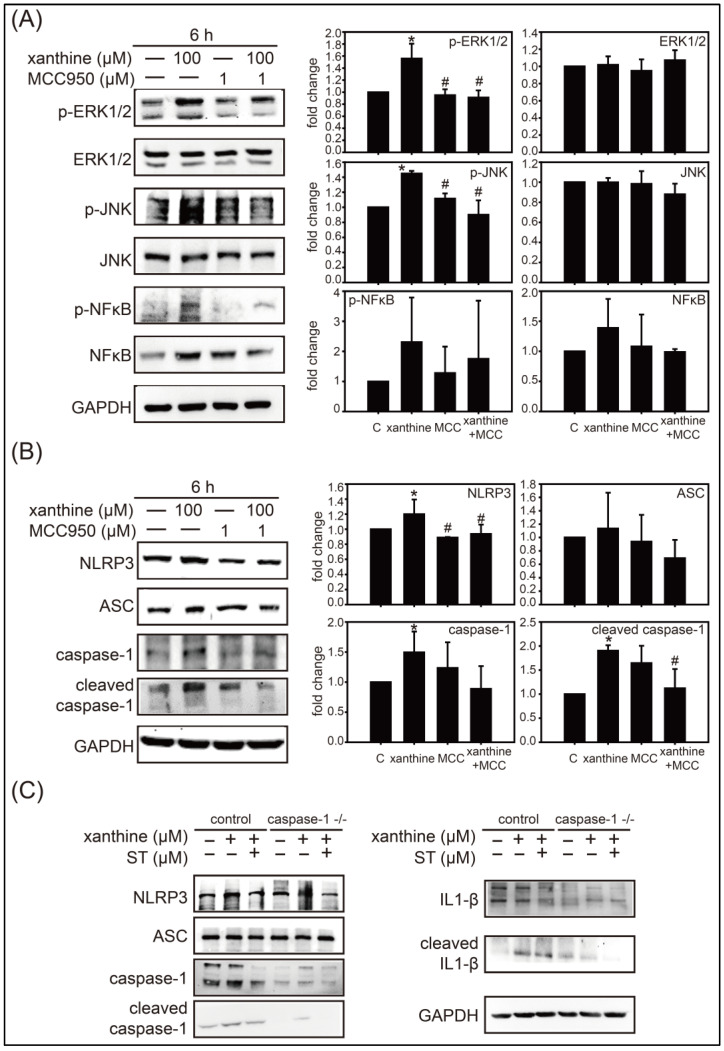
Downregulation of strictinin (ST) on signal transduction and NLRP3 inflammasome pathways. Western blot was conducted with cell lysates from AML12 cells after treatment with xanthine (100 μM) and/or inhibitor MCC950 (1 μM) for 6 h. Representative images and quantitation of the Western blot are shown for the protein expressions related to signal transduction pathway (**A**) and NLRP3 inflammasome pathway (**B**). NLRP3 inflammasome-related protein expressions were detected in caspase-1-silenced AML 12 cells (caspase-1 -/-) treated with xanthine and/or ST for 24 h (**C**). GAPDH was an internal control for Western blot analysis. Data are presented as mean ± SD from three independent experiments. The quantification data for Figure 3C is shown in Appendix A. The densitometry readings of Western blot analysis was shown in Appendix A. * *p* < 0.05 compared with control. # *p* < 0.05 compared with xanthine alone groups.

**Figure 4 biology-12-00329-f004:**
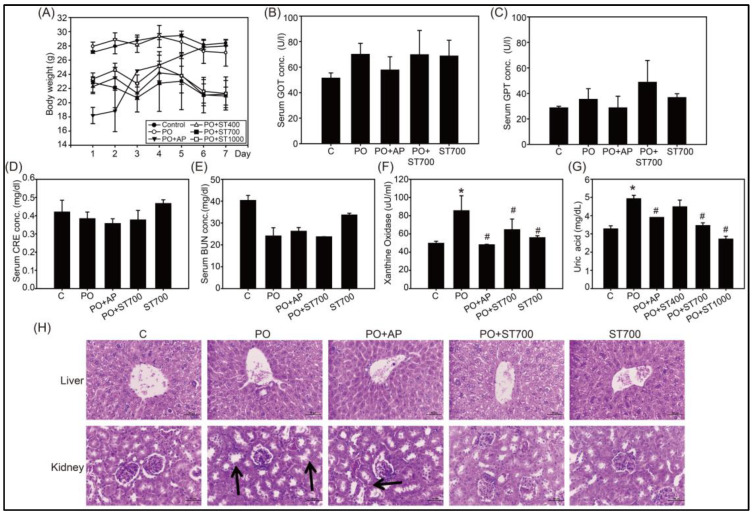
Effects of strictinin (ST) in PO-treated hyperuricemia C57BL/6 mice. PO-treated mice were supplemented with allopurinol (AP) or ST. Changes in body weight (**A**), glutamic oxaloacetic transaminase (GOT) (**B**), glutamic pyruvic transaminase (GPT) (**C**), creatinine (CRE) (**D**), blood urea nitrogen (BUN) (**E**), XOD activity (**F**), and serum uric acid production (**G**) were measured. Histopathological assessment of mice kidney and liver tissues are shown by representative images (**H**). Renal tubular dilation is indicated by arrows. Scale bar = 50 μm. * *p* < 0.05 compared with the control groups. # *p* < 0.05 compared with the PO-treated groups. Data are presented as mean ± SD from three independent experiments.

**Figure 5 biology-12-00329-f005:**
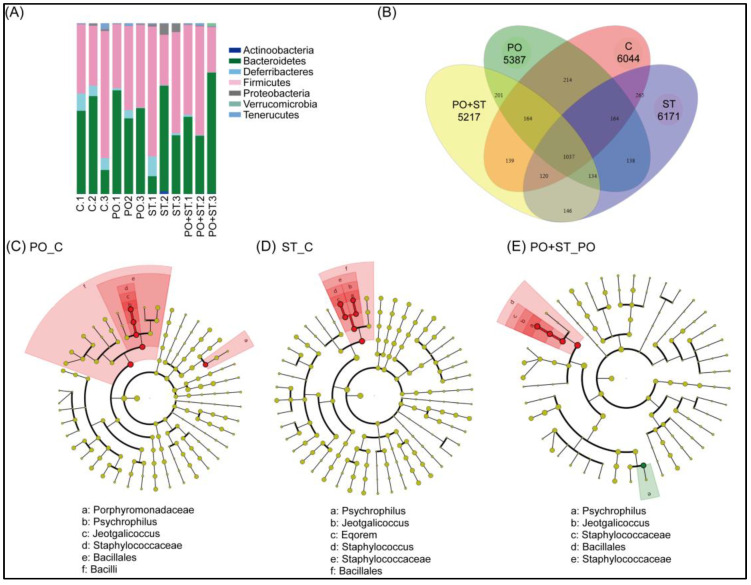
Analysis of mouse gut microbiota. Taxonomy composition bar plots indicate bacterial phyla in the control group (C.1–C.3), PO-treated group (PO.1–PO.3), strictinin-treated group (ST.1–ST.3), and PO+strictinin-treated group (PO+ST.1–PO+ST.3) (**A**). Venn diagram was created based on the Operational Taxonomic Unit (OTU), which showed the correlation of number of species between different treatment groups (**B**). The cladograms show the results of Linear Discriminant Analysis (LDA) of the significant difference in the abundances of gut microbiota between the PO-treated group and the control group (PO_C) (**C**), between the strictinin-treated group and the control group (ST_C) (**D**), and between the PO+strictinin-treated group and the PO-treated group (PO+ST_PO) (**E**) (n = 3 in each group).

**Figure 6 biology-12-00329-f006:**
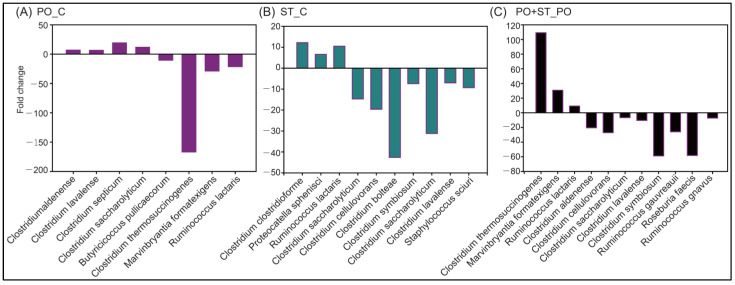
Metastats analysis of 16S rRNA operational taxonomic unit clustering using sequence similarity. Significant species-level changes were identified between the PO-treated group and the control group (PO_C) (**A**), between the strictinin-treated group and the control group (ST_C) (**B**), and between the PO+strictinin-treated group and the PO-treated group (PO+ST_PO) (**C**). (n = 3 in each group).

## Data Availability

The data presented in this study are available on request from the corresponding author.

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
