# Peer review of "Alleviation of Hyperuricemia by Strictinin in AML12 Mouse Hepatocytes Treated with Xanthine and in Mice Treated with Potassium Oxonate"

_biology, 2023, doi:10.3390/biology12020329_

Round 1

Reviewer 1 Report

To the authors

In this paper, the authors reported the effectiveness of strictinin, an abundant polyphenol in Pu’er tea, for preventing hyperuricemia. Strictinin reduced the xanthine oxidase activity, uric acid production, and inflammation in AML12 mouse hepatocytes treated with xanthine. Strictinin also reduced blood uric acid levels and enhanced xanthine oxidase activity in mice treated with potassium oxonate. This topic is interesting; however, several problems should be resolved prior to acceptance for publication in Biology, as shown below.

Major points

1.       In Fig. 1, 250 µM of ST treatment appears to reduce cell viability. Furthermore, 250 μM of ST treatment seems to increase the production of XO and UA, and these results raise concerns about the appropriateness of the dose of ST treatment. In Fig. 2, NLRP3 and Caspase-1 also appear to be upregulated by ST treatment.

2.       In Figure 4, serum Cre and BUN decreased due to the administration of PO. On the other hand, the administration of PO induced renal tubular damage in HE staining, and this result is contradictory. How was the renal pathological evaluation performed? For assessment of the adequacy of renal injury, it is necessary to investigate biomarkers of renal tubular disorders such as urinary Kim-1.

3.       The discussion section does not state the limitations of this study. In the future, it will be important for researchers interested in hyperuricemia to know the limitations of strictinin when using it. Therefore, the authors need to clarify the limitation(s) of this study.

Overall, this is a well written and interesting manuscript. However, I have concerns regarding the safety of ST treatment.

Reviewer 2 Report

Authors unrevealed the potential beneficial effects of natural polyphenol Strictinin abundantly found in Pu’er Tea, all the data described using in vitro cell culture model AML12 and in vivo mouse model developed with hyperuricemia upon potassium oxonate administration in diet. In addition to beneficial effects in liver and kidney, also play critical role in modulation of gut microbiome involved in pathogenesis and inflammation. The experimental design is well established and in vivo and in vitro experiments might need additional support to conclude the hypothesis. Authors need to address the concerns before the current manuscript accepted for publication in Biology.

Major concerns:

11)  Number of mice in each group is very limited (N=3),

22) In Figure 4, authors showed several markers related to liver and kidney injury. However, creatinine levels were unchanged between control and model (mice fed PO) Also BUN seems to be decreased significantly in the model. At this condition, why authors found the tubular dilation in the kidney? How does this relate to slight damage in kidney? What profound factors that could involved in minimal damage?

33) Cell culture studies showing the anti-inflammatory effect of Strictinin. Also, this natural compound has anti-oxidative effect. However, authors did show any markers related oxidative stress including ROS which can be induced by Xanthine.  

Round 2

Reviewer 1 Report

To the authors

Thank you for the opportunity to review the revised manuscript. The comments raised by the reviewer are well addressed.

Reviewer 2 Report

Thanks to authors for responses. All the comments and suggestions were addressed accordingly. The current version can accepted for manuscript in Biology. Congratulation to all the authors.